# Economic complexity of prefectures in Japan

**Abhijit Chakraborty** [1,2,3] *, **Hiroyasu Inoue**[1], **Yoshi Fujiwara**[1]

**1** Graduate School of Simulation Studies, The University of Hyogo, Kobe, Japan, **2** Advanced Systems Analysis, International Institute for Applied Systems Analysis (IIASA), Laxenburg, Austria, **3** Complexity Science Hub Vienna, Vienna, Austria

* abhiphyiitg@gmail.com

**Data Availability Statement:** The data for bipartite network is based on a survey done by Tokyo Shoko Research (http://www.tsr-net.co.jp/), one of the leading credit research agencies in Tokyo, and is supplied through the Research Institute of Economy, Trade and Industry. The data are not in

## Abstract

Every nation prioritizes the inclusive economic growth and development of all regions. However, we observe that economic activities are clustered in space, which results in a disparity in per-capita income among different regions. A complexity-based method was proposed by Hidalgo and Hausmann [PNAS 106, 10570-10575 (2009)] to explain the large gaps in per-capita income across countries. Although there have been extensive studies on countries' economic complexity using international export data, studies on economic complexity at the regional level are relatively less studied. Here, we study the industrial sector complexity of prefectures in Japan based on the basic information of more than one million firms. We aggregate the data as a bipartite network of prefectures and industrial sectors. We decompose the bipartite network as a prefecture-prefecture network and sector-sector network, which reveals the relationships among them. Similarities among the prefectures and among the sectors are measured using a metric. From these similarity matrices, we cluster the prefectures and sectors using the minimal spanning tree technique. The computed economic complexity index from the structure of the bipartite network shows a high correlation with macroeconomic indicators, such as per-capita gross prefectural product and prefectural income per person. We argue that this index reflects the present economic performance and hidden potential of the prefectures for future growth.

## Introduction

An important characteristic of the economy is that economic activities are heterogeneously distributed over geographic locations. For example, the "blue banana" region, which stretches from southeastern England through the Benelux countries, northern France and southwestern Germany to northeastern Italy, has a high level of income compared to other regions in Europe. Similar disparities in economic activity have also been observed at the national level. In Portugal, differences in development activities are observed between Lisbon and the north of the country and the center and the south of the country. Similar examples include Paris compared to the rest of France, northeastern Spain and Madrid compare to the south and west parts of Spain, and the southern versus northern UK. Moreover, Japan also follows this trend. Japanese prefectures such as Tokyo and Osaka are much more developed than rural prefectures such as Akita and Kagoshima [1, 2].

the public domain, but are commercially available. Data access requests for the TSR Company Profile Data File can be directed to the Tokyo Shoko Research, Ltd. (contact via telephone: +81 (0)3-6910-3142, or via fax: +81 (0)3-5221-0712). The data can be accessed by others in the same manner by which the authors obtained them. The gross prefectural product data, prefectural population data and prefectural income per person data for the year 2015 were retrieved from the Japanese government statistical portal site (https://www.e-stat.go.jp), and are in the public domain.

**Funding:** This research was supported by MEXT as Exploratory Challenges on Post-K computer (Studies of Multilevel Spatiotemporal Simulation of Socioeconomic Phenomena).

**Competing interests:** The authors have declared that no competing interests exist.

Hidalgo and Hausmann proposed a complexity-based method to analyze the structural properties of bipartite world trade networks to explain large gaps in per-capita income across countries [3, 4]. They quantitatively measured the complexity indices of the countries and their export products from the trade network, as these economic complexity indices are useful for explaining countries' performance. In a recent work, Mealy *et al.* showed [5] that the complexity index is equivalent to a spectral clustering algorithm, which divides a similarity graph into two parts. They have further shown that these indices are connected to various dimensionality reduction methods. Subsequently, Tacchella *et al.* introduced the fitness-complexity algorithm [6] based on the conceptual framework of Hidalgo and Hausmann to calculate intangible properties such as the fitness of countries and the complexity of export products from the structure of the world trade network. This method is very similar to the Google page rank method for directed networks and applicable to bipartite networks. In this algorithm, the fixed point of coupled nonlinear maps provides the fitness of countries and the complexity of products. The comparison of the complexity indices obtained by both methods [4, 6] with standard monetary indices presents an indication for potential future growth.

Economic complexity has traditionally been studied considering the structure of the bipartite world trade network [3, 4, 6–11]. Recently, economic complexity has been studied at the regional level for China [12], Brazil [10], Mexico [13], Italy [14], Spain [15], Australia [16], the US and the UK [5]. Most of these regional complexity studies are done at very coarse grain level. In case of China, the analysis is performed for 31 provinces with 2690 firms, which is a tiny fraction of all Chinese firms. The complexity analysis is performed at states level for Brazil, Mexico and Australia. The difference in our study is that it concerns supply-chain in which prefectures and industrial sectors are studied. We are looking at process of value added starting from a giant network of firms and by aggregating as a binary bipartite network of prefectures and industrial sectors. The investigation of the structure of bipartite networks of cities and their economic activities shows similarities with the nested ecological networks observed in mutualistic interactions between species [17]. These complexity methods have also been studied in regard to ecological networks [18]. The quantification of complexity is found to be useful for ranking active and passive species in ecological networks.

Japan has been one of the most diversified country in the sense of the products. Therefore, it is important to reveal that whether such diversity comes from regional structures. We use information about more than one million Japanese firms for this study. Similar Japanese firm-level data have been investigated in the past [19–25]. Past studies on these datasets mostly aimed at uncovering the structure and dynamics of the supply chain network and bank-firm credit network. However, the network that represents the interactions of firms with geographic locations has not yet been holistically studied. Here, we uncover the industrial sector complexity of prefectures in Japan from the structure of the bipartite network of prefectures and their economic activities. The bipartite network is based on basic information of more than one million Japanese firms. Using the locations of the firms and Japan Standard Industrial Classification, we aggregate the data as a bipartite network of prefectures and industrial sectors. The monopartite projection of the bipartite network presents a prefecture-prefecture network and sector-sector network. The similarities among prefectures and among industrial sectors are measured with these monopartite networks. Using the measured similarities, clustering among prefectures and sectors is shown with the minimal spanning trees (MSTs). By employing the economic complexity framework, we calculate the economic complexity index (ECI) for the prefectures, which exhibit a high correlation with macroeconomic indicators, per-capita gross prefectural product and prefectural income per person. Furthermore, we have checked the robustness of the economic complexity results using the fitness complexity method [6].

The rest of this paper is structured as follows. In the Data section, we provide the descriptions of the data. We explain the details of the methods in the Methods section. In the Results section, we present the results of our investigation, and in the Conclusions section, we present our conclusions.

## Data

Our data are based on a survey conducted by Tokyo Shoko Research (TSR), one of the leading credit research agencies in Tokyo, which was supplied to us by the Research Institute of Economy, Trade and Industry (RIETI). We use "TSR Kigyo Jouhou" (firm information), which contains basic financial information on more than one million firms. The dataset was compiled in July 2016. We only considered "active" firms that have information on employees and current year sales. The dataset contains $N = 1,033,518$ firms. These firms constitute a giant weakly connected component in the Japanese production network [21]. The industrial sectors are hierarchically categorized into 20 divisions, 99 major groups, 529 minor groups and 1,455 industries (Japan Standard Industrial Classification, November 2007, Revision 12). We aggregate the data as a bipartite network of prefectures ($P = 47$) and industrial sectors ($S = 91$). We exclude some of the industrial sectors from the 99 major groups of the industrial sector classification, as these sectors skew the analysis in the following way: the excluded sectors are manufacturers of petroleum and coal products, services incidental to the internet, financial product transaction dealers and future commodity transaction dealers, professional services, advertising services, and postal services. As these excluded sectors are only linked to Tokyo, the inclusion of these sectors in our analysis results in the largest value for the fitness of Tokyo, and the fitness of other prefectures become zero.

The bipartite network is represented by the binary matrix $M_{ps}$, where $M_{ps} = 1$ if the industrial sector $s$ has a significant amount of annual sales in prefecture $p$ and 0 otherwise. The Revealed Comparative Advantage (RCA) [26] is frequently used as a quantitative criterion to evaluate the relative dominance of a country, in the export of certain products by comparing it with the average export of those products. Recently, RCA has been measured from the ratio between the actual number of firms from an industry in a province and the average number of firms from that industry in that province [12]. Mealy *et al.* constructed a binary region-industry matrix based on the number of people employed in an industry in a region [5]. Here, we use annual sales of industrial sector $s$ in prefecture $p$ to measure the RCA, which is also a good indicator of the performance of a industrial sector. An industrial sector $s$ is said to have a significant amount of annual sales in prefecture $p$ if its revealed comparative advantage (RCA) is greater than or equal to unity.

The RCA is defined as

$$RCA_{ps} = \frac{\dfrac{w_{ps}}{\sum_s w_{ps}}}{\dfrac{\sum_p w_{ps}}{\sum_{p,s} w_{ps}}},$$

where $w_{ps}$ is the aggregated annual sales of industrial sector $s$ in prefecture $p$.

To explain the heterogeneity in prefectural economic activities, we have examined the relationship between economic complexity and certain macroeconomic factors characterizing a prefectural economy. In particular, we find relationships between economic complexity and per-capita gross prefectural product and with prefectural income per person. The gross prefectural production is the total amount of value added produced in the prefecture and is calculated by subtracting raw material costs and utility costs from the total amount of services

produced in the prefecture. Per-capita gross prefectural product is obtained by dividing the prefectural gross production by the prefectural population. Prefectural income is the sum of employee compensation, property income and business income. The prefectural income per person is obtained by dividing the prefectural income by the prefectural population. We collected the gross prefectural product data, prefectural population data and prefectural income per person data for the year 2015 from the Japanese government statistical portal site (https://www.e-stat.go.jp).

## Methods

### Method for measuring economic complexity

Hidalgo and Hausmann introduced the idea of economic complexity for countries and products that they export [3, 4]. Here, we apply the method to Japanese prefectures and their industrial sectors. The economic complexity index (ECI) of prefectures and product complexity index (PCI) of industrial sectors can be calculated using the following iterative equation:

$$k_{p,N} = \frac{1}{k_{p,0}} \sum_s M_{ps} k_{s,N-1} \tag{1}$$

$$k_{s,N} = \frac{1}{k_{s,0}} \sum_p M_{ps} k_{p,N-1}, \tag{2}$$

where $k_{p,0} = \Sigma_s M_{ps}$ and $k_{s,0} = \Sigma_p M_{ps}$. In network terms, $k_{p,1}$ and $k_{s,1}$ are known as the average nearest neighbor degree.

Substituting Eq (2) into Eq (1) obtains

$$k_{p,N} = \frac{1}{k_{p,0}} \sum_s M_{ps} \frac{1}{k_{s,0}} \sum_{p'} M_{p's} k_{p',N-2} \tag{3}$$

$$k_{p,N} = \sum_{p'} k_{p',N-2} \sum_s \frac{M_{ps} M_{p's}}{k_{p,0} k_{s,0}} = \sum_{p'} \widetilde{M_{pp'}} k_{p',N-2}, \tag{4}$$

where

$$\widetilde{M_{pp'}} = \sum_s \frac{M_{ps} M_{p's}}{k_{p,0} k_{s,0}} \tag{5}$$

Eq (4) is satisfied when $k_{p,N} = k_{p',N-2} = 1$, which is the eigenvector of $\widetilde{M_{pp'}}$ associated with the largest eigenvalue. Since this eigenvector is a vector with identical component values, it is not informative. The eigenvector associated with the second largest eigenvalue captures the largest amount of variance in the system. Therefore, we define the ECI as follows:

$$ECI = \frac{\vec{K} - <\vec{K}>}{stdev(\vec{K})}, \tag{6}$$

where $\vec{K}$ is the eigenvector of $\widetilde{M_{pp'}}$ associated with the second largest eigenvalue. $<\vec{K}>$ and $stdev(\vec{K})$ indicate the mean and standard deviation of the components of the eigenvector $\vec{K}$, respectively.

To further understand the matrix elements $\widetilde{M_{pp'}}$, one can write Eq (5) in the following way:

$$\widetilde{M_{pp'}} = \sum_s \frac{M_{ps}}{k_{p,0}} \frac{M_{p's}}{k_{s,0}} = \sum_s P(s|p)P(p'|s) = P(p'|p), \tag{7}$$

where $P(s|p) = M_{ps}/k_{p,0}$ is the conditional probability that any industrial sector $s$ is present in a given prefecture $p$, and $P(p'|s) = M_{p's}/k_{s,0}$ is the conditional probability that a particular industrial sector $s$ is present in any prefecture $p'$. From Eq (7), we can interpret $\widetilde{M_{pp'}}$ as the conditional probability of reaching $p'$ from $p$ through common industrial sectors.

Similarly, one can calculate the product complexity index (PCI) from the eigenvector associated with the second largest eigenvalue of the matrix:

$$\widetilde{M_{ss'}} = \sum_p \frac{M_{ps}M_{ps'}}{k_{p,0}k_{s,0}}. \tag{8}$$

## Fitness-complexity algorithm

Based on the conceptual framework of Hidalgo and Hausmann [3] and inspired by the Google page rank algorithm, Tacchella *et al.* introduced the fitness-complexity algorithm [6]. This method has been studied extensively in regard to countries and their export products [9–11]. Using this method, one can calculate the intangible properties such as the fitness of countries and the complexity of products. Here, we use this method to study Japanese industrial sector and prefecture relationships.

This method is based on the following three ideas. (i) The fitness of a prefecture is measured in terms of the diversity of the industrial sector set, weighted by the complexity of sectors. (ii) The more prefectures there are that have a particular industrial sector, the lower the complexity of the industrial sector. (iii) The upper bound of the complexity of an industrial sector must be dominated by the prefectures with the lowest fitness.

The above facts are mathematically represented by the following self-consistent iterative coupled equations with fitness $F_p$ of prefectures and complexity $Q_s$ of industrial sectors:

$$\widetilde{F}_p^{(n)} = \sum_s M_{ps}Q_s^{(n-1)},$$

$$\widetilde{Q}_s^{(n)} = \frac{1}{\sum_p M_{ps}\frac{1}{F_p^{(n-1)}}}, \tag{9}$$

with normalization in each step: $F_p^{(n)} = \frac{\tilde{F}_p^{(n)}}{<\tilde{F}_p^{(n)}>}$; $Q_s^{(n)} = \frac{\tilde{Q}_s^{(n)}}{<\tilde{Q}_s^{(n)}>}$. Here, $n$ represents any arbitrary iteration step.

The initial conditions are $\widetilde{Q}_s^{(0)} = \widetilde{F}_p^{(0)} = 1$ for all $p$ and $s$. The nature of the fixed point of the above equations depends on the structure of $M_{ps}$ [27].

We use the fitness-complexity method to check the robustness of the ECI for prefectures.

## Results

Bipartite network projection is a useful technique to compress information about bipartite networks. The bipartite network of prefectures and industrial sectors can be decomposed into two networks, namely, the network of prefectures and the network of industrial sectors.

## The network of prefectures

The projection network of prefectures is represented by the ($N_p \times N_p$) prefecture-prefecture matrix $P = MM^T$. The nondiagonal element $P_{pp'}$ corresponds to the number of industrial sectors that prefecture $p$ and $p'$ have in common. The diagonal element $P_{pp}$ corresponds to the number of industrial sectors belonging to prefecture $p$ and is a measure of the diversification of prefecture $p$. To quantify the competition among two prefectures, we can define the similarity matrix among prefectures as

$$\Theta_{pp'}^{P} = \frac{2 \times P_{pp'}}{P_{pp} + P_{p'p'}},$$

where $0 \leq \Theta_{pp'}^{P} \leq 1$. The values of $\Theta_{pp'}^{P}$ indicate a correlation between the industrial sectors of prefectures $p$ and $p'$.

We have investigated the interrelation between the different prefectures by considering how similar they are in terms of their industrial sectors. The MST is a widely used method to visualize the similarities between nodes. Given a set of nodes with a matrix specifying the similarity between them, the method of MST involves the following steps: (i) initially, an arbitrary node is set as a tree; (ii) the tree is grown with a link that has maximum similarity; and (iii) step (ii) is repeated until all nodes are merged with the tree. We have shown the clustering of prefectures by the MST in Fig 1. By visual inspection, we can observe that three different clusters on the tree consist of four prefectures of the Kanto, Chubu, and Kyushu regions. There is also a cluster of four prefectures of the Tohoku region and Hokkaido. Moreover, we observe various highly correlated pairs of geographically closely located prefectures, such as Ehime-Kochi, Niigata-Nagano, Okayama-Hiroshima, Mie-Wakayama, and Hyogo-Osaka. This finding indicates a strong similarity among the regional industries and also reflects the cooperative and competitive nature of the regional industries in Japan.

## The network of industrial sectors

The bipartite network can also be projected as a network of industrial sectors. Similar to the prefecture network, the industrial sector network is represented by the ($N_s \times N_s$) sector-sector matrix $S = M^T M$. The nondiagonal element $S_{ss'}$ corresponds to the number of prefectures having both sectors $s$ and $s'$. The diagonal element $S_{ss}$ corresponds to the number of prefectures having sector $s$, which is a measure of the ubiquity of sector $s$. The similarity matrix among the sectors can be defined as

$$\Theta_{ss'}^{S} = \frac{2 \times S_{ss'}}{S_{ss} + S_{s's'}},$$

where $0 \leq \Theta_{ss'}^{S} \leq 1$. $\Theta_{ss'}^{S} = 1$ indicates that whenever industrial sector $s$ is present in a prefecture, industrial sector $s'$ is also present.

Similar to prefectures, we show the clustering of industrial sectors using the MST in Fig 2. Most of the manufacturing industrial sectors, except for manufacturers of food, chemical products, ceramic products, and information and communication electronics, form a single cluster among themselves, which may indicate that one manufacturing industrial sector depends on other manufacturing industrial sectors. We also observe a cluster of the construction sector and a cluster consisting of agriculture, forestry, fisheries and manufacturers of food industrial sectors. However, other sectors are scattered on the tree, and clusters are formed by the mixed composition of industrial sectors. For example, we observe that wholesale and retail trade industrial divisions do not appear together; rather, they are scattered all over the tree.

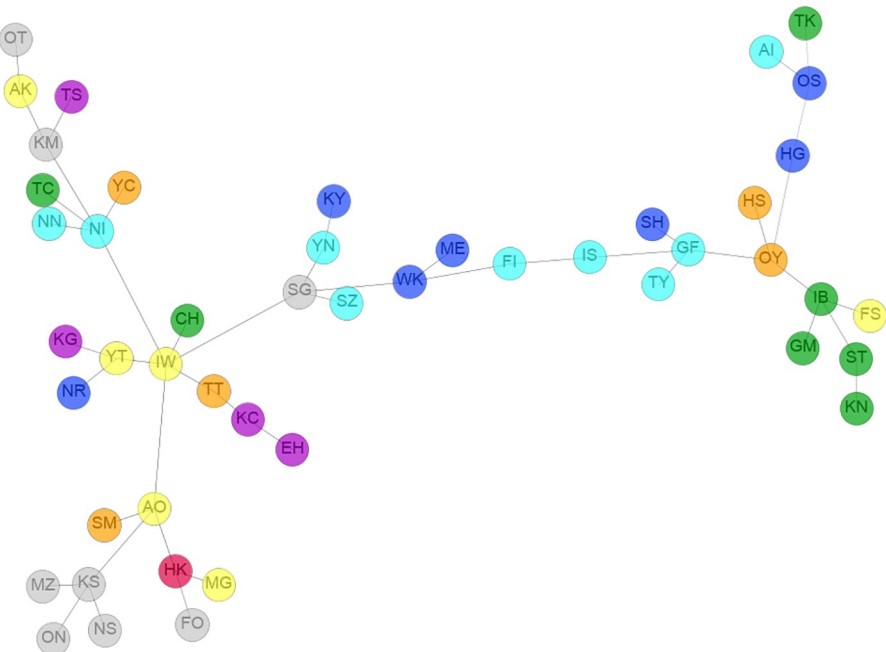

**Fig 1. The MST for prefectures.** The colors red, yellow, green, cyan, blue, orange, purple and light gray are used for the Hokkaido, Tohoku, Kanto, Chubu, Kansai, Chugoku, Shikoku and Kyushu regions, respectively. The codes of the prefectures are listed in S1 Table of S1 Appendix. The eight regions of Japan are shown in a map using the same color code in S1 Fig of S1 Appendix.

This analysis shows how industrial sectors are geographically similar and also indicates which industrial sectors are complementary.

## Economic complexity

We quantitatively measure the economic complexity of prefectures in Japan using the method of Hausmann and Hidalgo [3]. For the method details, see the Methods section. The industrial diversification of a prefecture is represented by $k_{p,0}$, and the ubiquity of its industrial sectors is indicated by $k_{p,1}$. We show the location of the prefectures in the space defined by $k_{p,0}$ and $k_{p,1}$ in Fig 3. $k_{p,0}$ and $k_{p,1}$ are slightly negatively correlated (Pearson correlation coefficient $r =$ −0.230 and p-value = 0.119), which indicate that many well diversified prefectures have ubiquitous industrial sectors. The two diversified prefectures, Tokyo and Osaka, have only less ubiquitous or highly specialized industrial sectors. Although Aichi is less diversified, it has highly specialized industrial sectors. This result is in stark contrast to the results found in the bipartite trade network of countries and their export products [3] and in the regional economic complexity of China [12], where a strong negative correlation is observed between these two quantities. The reported value of the Pearson correlation coefficient in the case of China's regional complexity is $r =$ −0.777, and the p-value is = $2.8 \times 10^{-7}$ [12].

The ECI is a quantitative measure of the complexity of a prefecture and a nonmonetary variable and can capture the economic development of a region [3, 4, 12]. For prefectures, we can compare the ECI with macroeconomic variables such as per-capita gross prefectural product and prefectural income per person. We show the relationship between the ECI and per-capita gross prefectural product and prefectural income per person in Fig 4. The ECI has a strong positive correlation with per-capita gross prefectural product (Pearson correlation coefficient $r = 0.661$ with a p-value = $4.2 \times 10^{-7}$) and prefectural income per person (Pearson correlation

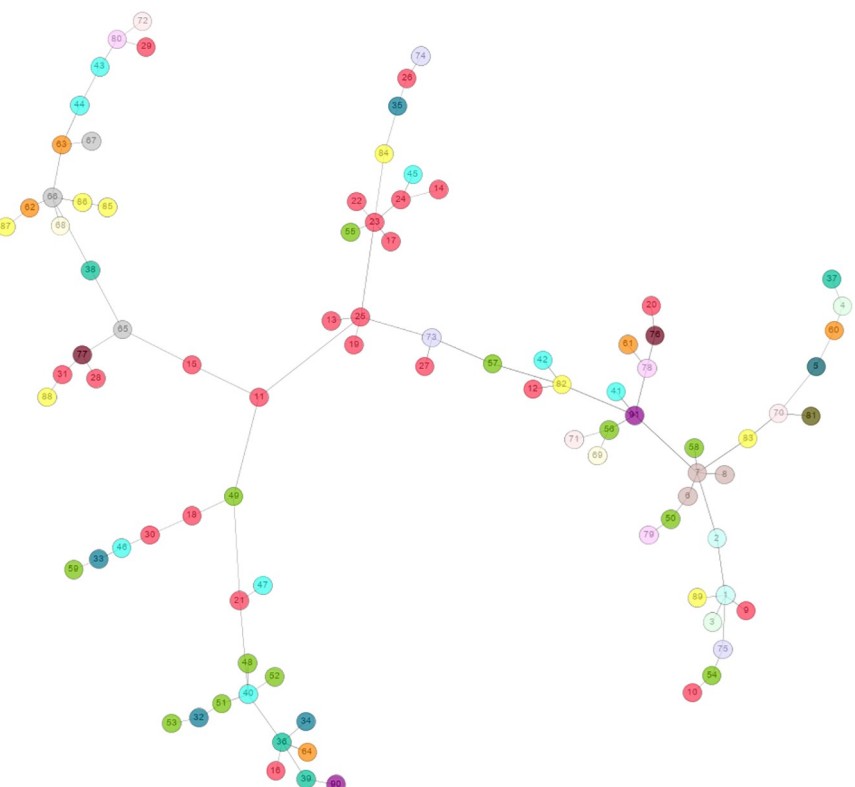

**Fig 2. The MST for industrial sectors.** Different colors represent nineteen divisions of industrial sectors. For example, red (ID: 9 to 31), light green (ID: 48 to 59), and brown (ID: 6 to 8) represent the manufacturing, wholesale and retail, and construction industrial sectors, respectively. The node IDs, sectors and divisions are given in S2 and S3 Tables of S1 Appendix.

coefficient $r = 0.668$ with a p-value $= 9.0 \times 10^{-8}$). Following [28], we can argue that the correlation between the macroeconomic factors and the ECI is observed because income growth rates are similar for prefectures with similar industrial sectors. An exponential fit to the data reflects the expected values of per-capita gross prefectural product and prefectural income per person at their level of economic complexity. The deviations in real per-capita gross prefectural product and prefectural income per person data from the expected values are informative and provide an indication of the economic performance of the prefectures. Prefectures such as Osaka, Kanagawa, Hyogo, Fukuoka, and Okinawa, appearing below the expected values of per-capita gross prefectural product and prefectural income per person, may have the potential to more quickly grow in the future. An interpretation of the above results for the regions in Japan is given in the section "the average prefectural economic complexity of regions in Japan" of S1 Appendix.

## Robustness of the ECI using the fitness-complexity algorithm

To check the robustness of the ECI, we compare it with the results obtained using the fitness-complexity method [6]. For detailed descriptions of the method, see the Methods section. The convergence properties of the algorithm depend on the structure of $M_{ps}$ [27]. We investigate the triangular structure of binary matrix $M_{ps}$ by ordering the rows and columns according to their fitness complexity rank. The structure of the ordered $M_{ps}$ in Fig 5(a) shows that the diagonal line does not pass through the vacant region, which ensures that the fitness values of the

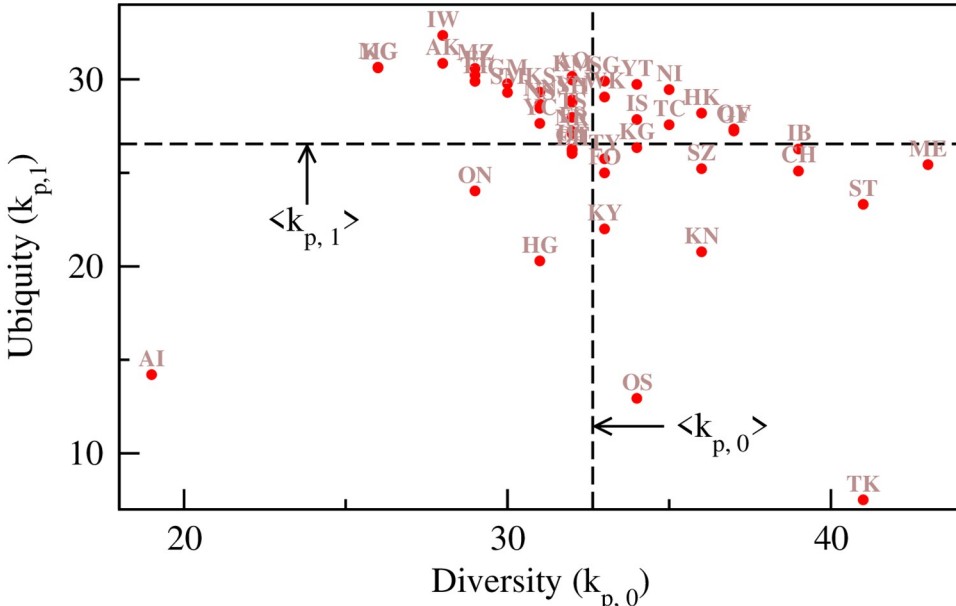

**Fig 3. Positions of the prefectures in the $k_{p,0} - k_{p,1}$ plane.** The diagram is divided into 4 quadrants, defined by the empirically observed averages $\langle k_{p,0} \rangle$ and $\langle k_{p,1} \rangle$.

prefecture and complexity values of the industrial sectors will converge to nonzero fixed values with iterations [27]. We indeed observe that the evolution of the fitness values of the prefectures reaches fixed nonzero values with iterations, as shown in Fig 5(b).

As seen from Fig 6, similar to the ECI, the fitness of the prefectures also shows a strong positive correlation with the per-capita gross prefectural product (Pearson correlation coefficient $r = 0.742$ and a p-value $= 2.3 \times 10^{-9}$) and prefectural income per person (Pearson correlation coefficient $r = 0.746$ and a p-value $= 1.8 \times 10^{-9}$). Here, we also observe that prefectures such as Osaka, Kanagawa, Hyogo, Fukuoka, and Okinawa appear below the expected values of the per-capita gross prefectural product and prefectural income per person. These prefectures may have the potential to more quickly grow in the future.

The ECI method and fitness complexity method obtain quite similar results. Comparisons of the ranking of the prefectures and industrial sectors by the two methods are listed in S2 and

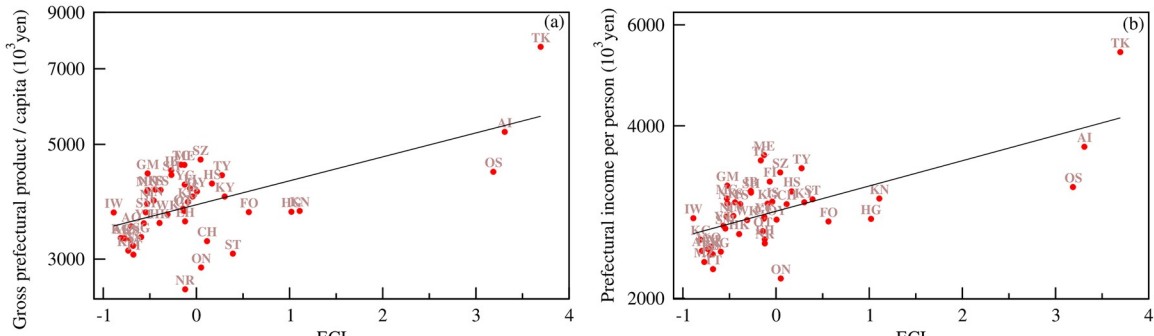

**Fig 4.** Variation in (a) per-capita gross prefectural product and (b) prefectural income per person in 2015 with the ECI. The straight lines in both plots represent an exponential fit to the data, indicating the expected values of the per-capita gross prefectural product and prefectural income per person.

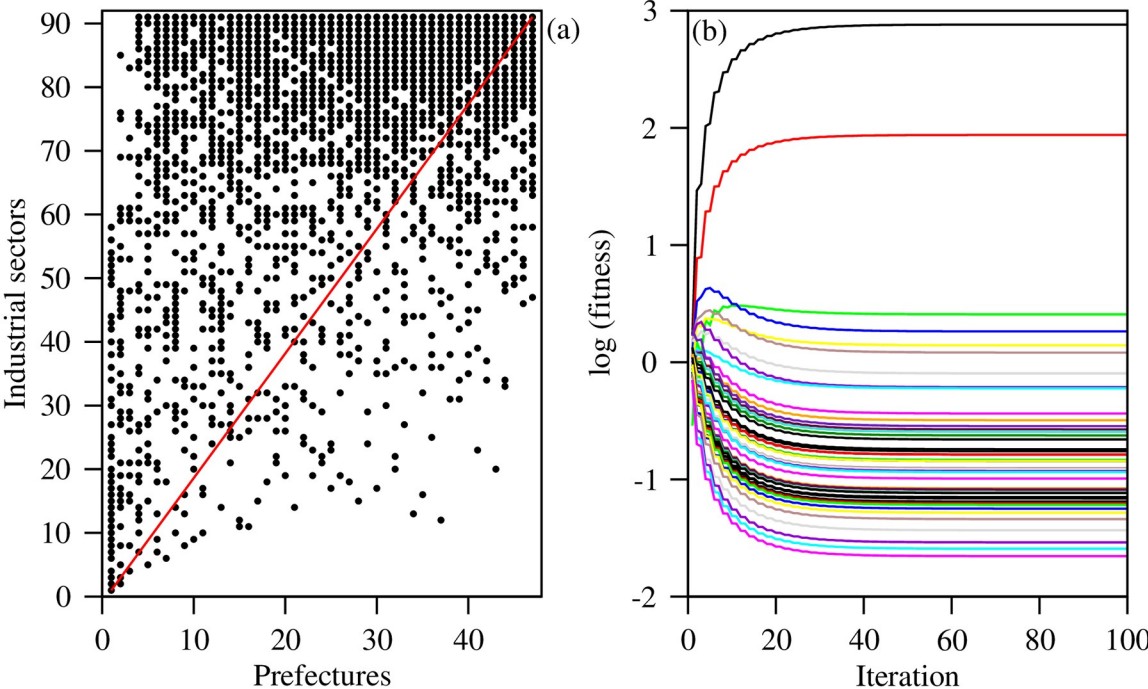

**Fig 5.** (a) Triangular structure of the ordered M matrix. (b) The evolution of the fitness values of the prefectures with iterations. The black dots in (a) represent that the industrial sector is present in the associated prefecture. There are a total of 47 curves in (b), and each of them represents the evolution of fitness values of a prefecture.

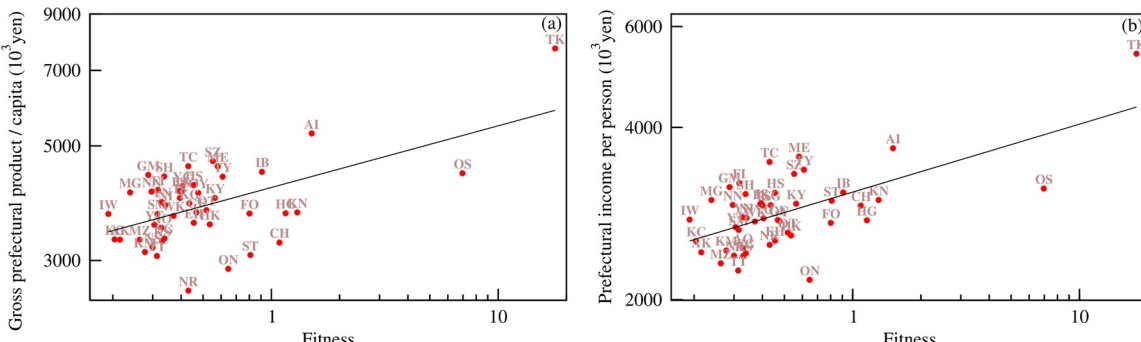

**Fig 6.** Variation in (a) per-capita gross prefectural product and (b) prefectural income per person in 2015 with fitness. The straight lines in both plots represent power law fit to the data, indicating expected values of the per-capita gross prefectural product and prefectural income per person.

S3 Tables of S1 Appendix, reflecting the fact that the nonmonetary variables ECI and fitness are good nonmonetary indicators for assessing the performance of a prefecture.

## Conclusions

We have studied the interactions of economic activities with prefectures in Japan using information on one million firms. The economic relation between prefectures shows that geographically close prefectures are cooperative and competitive. The interrelationship between industrial sectors shows the interdependence among them. The clustering of industrial sectors

further shows that the clusters are formed by diverse industrial sectors, except the manufacturing and construction sectors. We have observed that most of the diversified Japanese prefectures have ubiquitous industrial sectors, which is very different from the case of China [12] and in the international trades of countries [3]. The economic complexity measured by the nonmonetary variables, ECI and fitness for the prefectures shows a high correlation with macroeconomic indicators, such as per-capita gross prefectural product and prefectural income per person. These nonmonetary variables are very useful for understanding the economic activities in a prefecture. Our study will be helpful for understanding the economic health of industries in a region. We have studied economic complexity of prefectures in Japan based on the binary bipartite matrix. In the future, it will be interesting to see if one gets more valuable insights using a weighted matrix. Further studies on the dynamic evolution of economic complexity [29] in industrial sectors can predict the macroeconomic indicators for a prefecture.

## Supporting information

**S1 Appendix. Appendix to the manuscript.**
(PDF)

## Acknowledgments

We acknowledge computational resources IDs: hp160259, hp170242, hp180177, and hp190148.

## Author Contributions

**Conceptualization:** Abhijit Chakraborty.

**Data curation:** Abhijit Chakraborty.

**Formal analysis:** Abhijit Chakraborty.

**Funding acquisition:** Hiroyasu Inoue, Yoshi Fujiwara.

**Investigation:** Abhijit Chakraborty.

**Methodology:** Abhijit Chakraborty.

**Visualization:** Abhijit Chakraborty.

**Writing – original draft:** Abhijit Chakraborty.

**Writing – review & editing:** Abhijit Chakraborty, Hiroyasu Inoue, Yoshi Fujiwara.

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
