## [Decision Letter · Decision Letter 0]

21 Apr 2020

PONE-D-20-04110

Economic complexity of prefectures in Japan

PLOS ONE

Dear Dr. Chakraborty,

Thank you for submitting your manuscript to PLOS ONE. After careful consideration, we feel that it has merit but does not fully meet PLOS ONE’s publication criteria as it currently stands. Therefore, we invite you to submit a revised version of the manuscript that addresses the points raised during the review process.

We would appreciate receiving your revised manuscript by Jun 05 2020 11:59PM. To enhance the reproducibility of your results, we recommend that if applicable you deposit your laboratory protocols in protocols.io, where a protocol can be assigned its own identifier (DOI) such that it can be cited independently in the future. For instructions see: http://journals.plos.org/plosone/s/submission-guidelines#loc-laboratory-protocols

We look forward to receiving your revised manuscript.

Kind regards,

Hocine Cherifi

Academic Editor

PLOS ONE

Reviewers' comments:

Reviewer's Responses to Questions

**Comments to the Author**

1. Is the manuscript technically sound, and do the data support the conclusions?

Reviewer #1: Yes

Reviewer #2: Partly

Reviewer #3: Yes

2. Has the statistical analysis been performed appropriately and rigorously? 

Reviewer #1: Yes

Reviewer #2: Yes

Reviewer #3: Yes

3. Have the authors made all data underlying the findings in their manuscript fully available?

Reviewer #1: No

Reviewer #2: No

Reviewer #3: Yes

4. Is the manuscript presented in an intelligible fashion and written in standard English?

Reviewer #1: Yes

Reviewer #2: Yes

Reviewer #3: Yes

5. Review Comments to the Author

Reviewer #1: In their articles, the Authors analyze the economic complexity of the Japan industrial sector, with the aim of relating the healthy of the economic system with the geographical division of the considered region. In particular, they analyze the different prefectures that are present in Japan, firstly considering how similar are the industrial sectors that can be found in each of the prefectures. Then, they apply a series of published methods that are ad-hoc adapted for their case study to evaluate the economic health of each of the prefectures. The analysis is deeply done, and it can be followed even from readers, like me, that are not expert in this specific research field (a thing that is valuable for a broad impact journal like PlosOne).

I have three minor criticisms/suggestions, that in my opinion would greatly improve the paper:

1. To allow an easier understanding of the results even to not Japanese people, I suggest the Authors to add at list a figure in which there is the Japan geographical map divided into the different prefectures. The best, would be to present each of the obtained results suitably color coding a map: this would be very easy, for example, for Figs 1 and 2 (you just have to color the founded clusters of the tree and put the figure side of the tree...). A step in this direction would be very interesting also for the other figures, even if less obvious.

2. While the study is based on the geographical unit of the prefecture, it would be interesting to give an insight on the impact of the regional division. How the obtained results relate with the divisions in regions of the prefectures?

3. Finally, I think that the work the authors presented at the Complex Network conference that has exactly the same title should be properly referenced in the paper.

Reviewer #2: The authors apply the economic complexity methodology to the bipartite network of economic sectors and Japanese prefectures. Even if I am sympathetic with respect to the methodology, I do not think that the manuscript can be published in the present form. The authors could try to deeply revise it, trying to add economical and or theoretical investigations.

The main problem is the scientific contribution of this manuscript. It is stated in the abstract that "studies on economic complexity at the regional level are lacking", quite in contradiction with lines 32-33 "economic complexity has been studied at the regional level for China [12], Brazil [10], the US and the UK [5]", and I could add Mexico [1], Italy [2], Spain[3], and even Australia[4]. All these studies follow more or less the same route: first, a database is obtained with the export structure (i.e., different products) of each sub-national entity; then some normalization is performed (RCA); then the ECI or the Fitness or the Product Space algorithm is applied.

I believe that publishing the same exercise with different data is not very interesting from a scientific point of view. This analysis can be of some utility for Japanese policymakers, but does not add much to these known methodologies and also the economic meaning is not much discussed, for instance, with respect to more standard economical approaches. In conclusion, I find the data interesting but I think the authors should make a large effort to provide some better contribution to the literature.

Moreover, other important methodological issues are present:

- Some sectors are excluded from the analysis in a quite arbitrary way (as stated, because they are only linked to Tokyo, and this results in zero fitness for the other prefectures). Whether this is a data or a methodological problem, it should be clearly stated and discussed.

- The economic complexity methodology is usually applied to export data. Here RCA is computed from annual sales, which include internal and external production in a highly biased way. Both the use of sales instead of export and RCA should be motivated, possibly with some references.

- From Figures 4 and 6 it emerges that all the correlations are driven by Tokyo. What happens if it is removed?

Minor issues:

- Lines 101-102 are inaccurate. k_{p,N} is not the diversity of prefecture, as incorrectly stated, but the ECI. The diversity is k0. The same applies to sectors.

- The mathematical derivation in lines 104-125 is well known in the literature (ref. [7] and [24] of the paper), so it is quite useless.

- Also, the (quite elementary) similarity measures for both prefectures (pag.5) and sectors (page 6) is from [7].

- On lines 188-189 it is stated that "k p,0 and k p,1 are slightly negatively correlated (Pearson correlation coefficient r = −0.230 and p-value = 0.119)", which I find rather surprising. These iterative methods are supposed to converge to some "real value" of the economic complexity, providing better and better assessments as the iteration procedure goes on, so I would expect them to be highly correlated.

[1] Chávez, J. C., Mosqueda, M. T., & Gómez-Zaldívar, M. (2017). Economic complexity and regional growth performance: Evidence from the Mexican Economy. Review of Regional Studies, 47(2), 201-219.

[2] Basile, R., Cicerone, G., & Iapadre, L. (2019). Economic complexity and regional labor productivity distribution: evidence from Italy.

[3] Balsalobre, S. J. P., Verduras, C. L., & Lanchas, J. D. (2017). Measuring the Economic Complexity at the sub-national level using international and interregional trade.

[4] Reynolds, C., Agrawal, M., Lee, I., Zhan, C., Li, J., Taylor, P., ... & Roos, G. (2018). A sub-national economic complexity analysis of Australia’s states and territories. Regional Studies, 52(5), 715-726.

Reviewer #3: The paper is technically sound and the database is fantastic.

The statistical analysis is also adequate but more could be done with such a detailed database

The paper is also well written in English

The authors have made clear where the data was got from but I am not sure if they can share the detailed information on the firms. Could you please clarify this issue?

For more detailed comments please see my report.

6. PLOS authors have the option to publish the peer review history of their article (what does this mean?). If published, this will include your full peer review and any attached files.

Reviewer #1: No

Reviewer #2: No

Reviewer #3: No

---

## [Author Response · Author response to Decision Letter 0]

9 Jul 2020

Detailed responses to the comments by the Referees on manuscript PONE-D-20-04110 is given in "Response to Reviewers" document.

---

## [Decision Letter · Decision Letter 1]

30 Jul 2020

PONE-D-20-04110R1

Economic complexity of prefectures in Japan

PLOS ONE

Dear Dr. Chakraborty,

Thank you for submitting your manuscript to PLOS ONE. After careful consideration, we feel that it has merit but does not fully meet PLOS ONE’s publication criteria as it currently stands. Therefore, we invite you to submit a revised version of the manuscript that addresses the points raised during the review process.

We look forward to receiving your revised manuscript.

Kind regards,

Hocine Cherifi

Academic Editor

PLOS ONE

Reviewers' comments:

Reviewer's Responses to Questions

**Comments to the Author**

1. If the authors have adequately addressed your comments raised in a previous round of review and you feel that this manuscript is now acceptable for publication, you may indicate that here to bypass the “Comments to the Author” section, enter your conflict of interest statement in the “Confidential to Editor” section, and submit your "Accept" recommendation.

Reviewer #1: All comments have been addressed

Reviewer #3: (No Response)

2. Is the manuscript technically sound, and do the data support the conclusions?

Reviewer #1: Yes

Reviewer #3: Yes

3. Has the statistical analysis been performed appropriately and rigorously? 

Reviewer #1: Yes

Reviewer #3: Yes

4. Have the authors made all data underlying the findings in their manuscript fully available?

Reviewer #1: No

Reviewer #3: Yes

5. Is the manuscript presented in an intelligible fashion and written in standard English?

Reviewer #1: Yes

Reviewer #3: Yes

6. Review Comments to the Author

Reviewer #1: I'm completely satisfied with the Authors' paper revision, even if most of my comments have been answered in the SI. In my opinion, given also the Authors' reply, some figure should be moved in the main text, but I leave the Authors do what they think is the best.

Reviewer #3: I made a question in my previous revision but I am not satisfied with the answer. The authors should elaborate a lot more in their response and include it in the paper. I am referring to the question and reply below:

3) Is there a specific reason for the authors to use the unweighted version of the bipartite

network? I think that the weighted version can be quite useful in order to put monetary value to

the metrics, even for the projections the monetary value of the similarity can be quite useful as

well as a possible centrality study.

The algorithm that measures economic complexity index and product complexity index is

applied on a binary bipartite matrix [PNAS 106(26),10570 (2009)]. This is the reason we have

used the unweighted version of the bipartite network.

7. PLOS authors have the option to publish the peer review history of their article (what does this mean?). If published, this will include your full peer review and any attached files.

Reviewer #1: No

Reviewer #3: No

---

## [Author Response · Author response to Decision Letter 1]

4 Aug 2020

Detailed responses to the comments by the Referees on manuscript PONE-D-20-04110R1 (a copy of "Response to Reviewers" document):

We thank the Referees for their encouraging and thoughtful remarks as well as for their helpful suggestions. We reproduced all of their points below (in italics) together with our responses (in non-italics) to all points of criticism and suggestions. 

The changes in the revision result mostly from their suggestions and are therefore listed as well.

Reviewers' comments:

Reviewer's Responses to Questions Comments to the Author

1. If the authors have adequately addressed your comments raised in a previous round of review and you feel that this manuscript is now acceptable for publication, you may indicate that here to bypass the “Comments to the Author” section, enter your conflict of interest statement in the “Confidential to Editor” section, and submit your "Accept" recommendation.

Reviewer #1: All comments have been addressed Reviewer #3: (No Response)

2. Is the manuscript technically sound, and do the data support the conclusions?

Reviewer #1: Yes Reviewer #3: Yes

3. Has the statistical analysis been performed appropriately and rigorously? Reviewer #1: Yes

Reviewer #3: Yes

 4. Have the authors made all data underlying the findings in their manuscript fully available?

Reviewer #1: No

Reviewer #3: Yes

We have already provided a data availability statement which Plos one has approved by email communication with us on July 9, 2020. We share the statement below for your reference.

‘’The data for bipartite network is based on a survey done by Tokyo Shoko Research (http://www.tsr-net.co.jp/), one of the leading credit research agencies in Tokyo, and is supplied through the Research Institute of Economy, Trade and Industry. The data are not in the public domain, but are commercially available. Data access requests for the TSR Company Profile Data File can be directed to the Tokyo Shoko Research, Ltd. (contact via telephone: +81 (0)3-6910-3142, or via fax: +81 (0)3-5221-0712). Therefore, the data can be accessed by others in the same manner by which the authors obtained them.’’

5. Is the manuscript presented in an intelligible fashion and written in standard English?

Reviewer #1: Yes Reviewer #3: Yes

6. Review Comments to the Author

Reviewer #1: I'm completely satisfied with the Authors' paper revision, even if most of my comments have been answered in the SI. In my opinion, given also the Authors' reply, some figure should be moved in the main text, but I leave the Authors do what they think is the best.

We are pleased to know that our responses were helpful to address the reviewer’s concerns. We kept these figures in the SI such that readers do not divert from the main context.

Reviewer #3: I made a question in my previous revision but I am not satisfied with the answer. The authors should elaborate a lot more in their response and include it in the paper. 

I am referring to the question and reply below:

3) Is there a specific reason for the authors to use the unweighted version of the bipartite network? I think that the weighted version can be quite useful in order to put monetary value to the metrics, even for the projections the monetary value of the similarity can be quite useful as well as a possible centrality study.

The algorithm that measures economic complexity index and product complexity index is applied on a binary bipartite matrix [PNAS 106(26),10570 (2009)]. This is the reason we have used the unweighted version of the bipartite network.

We understand the reviewer's concern but the original economic complexity is defined as binary networks. Since we can compare our findings with results of the original economic complexity, we employ the original economic complexity. In addition, the expansion of the definition to the weighted networks can be another work and is beyond our scope.

Changes made in the manuscript:

At line 278-280: “We have studied economic complexity of prefectures in Japan based on the binary bipartite matrix. In the future, it will be interesting to see if one gets more valuable insights using a weighted matrix.”

To conclude, we again thank all the reviewers for providing us with the opportunity to clarify these issues. We have updated our manuscript according to the points discussed above. We hope that the changes we have made to the manuscript address the reviewers’ concerns.

---

## [Editor Report · Decision Letter 2]

10 Aug 2020

Economic complexity of prefectures in Japan

PONE-D-20-04110R2

Dear Dr. Chakraborty,

We’re pleased to inform you that your manuscript has been judged scientifically suitable for publication and will be formally accepted for publication once it meets all outstanding technical requirements.

Kind regards,

Hocine Cherifi

Academic Editor

PLOS ONE